# Quality Assessment of High-Speed Motion Blur Images for Mobile Automated Tunnel Inspection

**DOI:** 10.3390/s25123804

**Published:** 2025-06-18

**Authors:** Chulhee Lee, Donggyou Kim, Dongku Kim

**Affiliations:** Department of Geotechnical Engineering Research, Korea Institute of Civil Engineering and Building Technology (KICT), Goyang 10223, Republic of Korea; lch@kict.re.kr (C.L.);

**Keywords:** high-speed motion blur images, image quality assessment, modulation transfer function, blurred edge width, mobile tunnel scanning system

## Abstract

This study quantitatively evaluates the impact of motion blur—caused by high-speed movement—on image quality in a mobile tunnel scanning system (MTSS). To simulate movement at speeds of up to 70 km/h, a high-speed translational motion panel was developed. Images were captured under conditions compliant with the ISO 12233 international standard, and image quality was assessed using two metrics: blurred edge width (BEW) and the spatial frequency response at 50% contrast (MTF50). Experiments were conducted under varying shutter speeds, lighting conditions (15,000 lx and 40,000 lx), and motion speeds. The results demonstrated that increased motion speed increased BEW and decreased MTF50, indicating greater blur intensity and reduced image sharpness. Two-way analysis of variance and *t*-tests confirmed that shutter and motion speed significantly affected image quality. Although higher illumination levels partially improved, they also occasionally led to reduced sharpness. Field validation using MTSS in actual tunnel environments demonstrated that BEW and MTF50 effectively captured blur variations by scanning direction. This study proposes BEW and MTF50 as reliable indicators for quantitatively evaluating motion blur in tunnel inspection imagery and suggests their potential to optimize MTSS operation and improve the accuracy of automated defect detection.

## 1. Introduction

In South Korea, tunnels for railways, subways, and roads have been constructed to maximize land use. Over time, their performance deteriorates due to structural and environmental factors, leading to cracks, water leakage, and exposed rebar. Such damage can worsen because of inadequate maintenance and substandard repairs [1]. Notable incidents caused by poor maintenance include the 2006 ceiling collapse of the Big Dig tunnel in Boston (USA), the 2012 ceiling collapse of the Sasago tunnel in Japan, and the 2019 concrete lining collapse in the Berté tunnel on Italy’s A26 highway [2,3,4]. Proper maintenance, such as structural health monitoring (SHM), is essential for preventing such accidents, slowing deterioration, and sustaining tunnel performance [5]. Currently, tunnel inspections rely on visual assessments by qualified engineers to evaluate damage. Damage details and locations are typically mapped on the tunnel face, and repair methods are selected accordingly. However, more efficient and safer inspection methods are required owing to the low reliability of visual inspections, rising labor costs, and hazardous conditions [6].

Computer vision (CV) has been applied to automate inspections and manual analyses, becoming a key tool in civil engineering monitoring [7]. The use of deep learning (DL) for detecting surface damage in civil structures is increasing [8]. DL automatically extracts image features for damage detection, serving as an aid rather than a replacement for engineers in data collection and result interpretation [9]. Due to restrictions on public access to infrastructure data, most DL-based surface damage research relies on open datasets. For condition assessment, damage information is classified, localized, segmented, and quantified [10]. In tunnel maintenance, high precision is required—such as detecting cracks smaller than 0.3 mm—since repair methods depend on this threshold. Achieving such accuracy requires domain-specific datasets that consider defect types, environmental conditions, materials, and structural factors. Ultimately, the quality of training image data strongly influences surface damage detection performance.

A mobile tunnel scanning system (MTSS) with cameras was introduced in the early 2000s to automate tunnel inspections. However, existing systems have captured images without accounting for quality factors such as low resolution, motion blur, and noise, often overlooking environmental differences across tunnels. Image processing methods (IPMs) have been employed to detect damage and address site-specific environmental variations.

Cracks on the surface of concrete structures in tunnels can be characterized by two main features: they are thinner and distinct from other texture patterns in the structural form, and have low brightness. Therefore, IPM detection algorithms require structural specifications to extract dark objects from a bright background [11]. Crack detection techniques involving IPM in tunnel scanning systems include morphological Gabor filters [12], hat transforms [13], image fusion [14], Bayesian classifiers [15], and wavelet approaches [16]. These detection techniques detect cracks using grayscale images [17]. Additionally, with the increasing shooting speed of tunnel scanning systems, the volume and complexity of collected image data increase. This leads to the over-extraction of features by existing image processing algorithms, such as threshold segmentation [18], edge detection [19], and wavelet transforms [17]. Consequently, this results in time-consuming and complex data processing and analysis, often yielding inaccurate results [20]. Mobile DL-based tunnel scanning systems have been developed because of the need for accurate and effective analysis of collected image data.

Recently, research has been conducted using various CV and DL methods for SHM to detect surface damage to structures [21,22]. DL analysis to obtain high-accuracy and precision data required for SHM is more influenced by the quality of raw image data, such as pixel size, quality, and quantity, than the algorithm. Increased noise and motion blur in images captured during movement reduce the quality of the convolutional neural network (CNN)-based results [23]. Image quality variations can lead to inaccurate damage detection in public infrastructure like tunnels owing to environmental effects and low-quality training image data [24]. Ensuring high-quality image data enables CNN to achieve greater accuracy and precision in reproduction values. However, comprehensive research on the objective evaluation of image quality from MTSSs is still lacking, particularly considering factors affecting image quality, such as low resolution and motion blur. Furthermore, research on optimizing MTSS that collects high-quality raw image data to overcome data-related limitations to DL-based surface damage detection performance is also lacking.

This study aims to quantitatively analyze the quality of raw images captured by cameras in the MTSS. It evaluates the performance of currently operating systems and examines the potential of image quality assessment (IQA) metrics as foundational data for developing advanced future equipment. To this end, this study develops a translational moving panel device that simulates indoor high-speed movement at 70 km/h, considering the driving directionality of MTSS. An indoor testing environment was constructed based on the international standard ISO 12233, which is used for analyzing camera resolution performance. Motion-blurred images were captured at various physical movement speeds under different camera exposure settings and lighting conditions. These images were comparatively analyzed using IQA metrics. Additionally, the applicability of these IQA metrics was evaluated using a currently operational MTSS, aiming to suggest a direction for future image quality management of the MTSS.

## 2. Related Research

### 2.1. Camera-Based Tunnel Scanning Systems for Automation in Inspection

Regarding tunnels with continuous sections of the same shape and size, automated inspection can be achieved by scanning the entire concrete lining using imaging devices such as cameras. Mobile camera-based tunnel scanning systems can be broadly categorized into those designed for railways and those for roads. Railway tunnel scanning systems have been continuously developed alongside advancements in camera image sensor technology and typically capture images at 5–10 km/h using trolley systems [25,26,27,28,29,30,31,32,33,34]. Railway tunnel inspections face temporal limitations because they must be conducted during non-operational hours. Low-speed inspection equipment is used because of the challenges of setting up high-speed movement devices within a short timeframe.

Conversely, road tunnels have fewer constraints compared with railway tunnels. The speed of vehicles can be autonomously controlled or adjusted for inspections by controlling or partially closing certain lanes. Additionally, road tunnels can be inspected anytime, day or night, providing ample inspection time. Typically, road tunnel scanning systems can be mounted on large vehicles equipped with cameras and lighting, facilitating transportation ease and offering excellent adaptability for tunnel inspections [35]. These systems are being developed to detect various types of damage from images captured at 40–80 km/h [36,37,38,39,40,41,42,43,44].

MTSSs and DL technologies are being developed because of the increasing need for more accurate and effective detection of damage such as cracks from image data. Xue et al. [45] proposed and designed the Faster Region-based CNN (Faster RCNN) for crack detection in images acquired using MTI-100 railway tunnel scanning equipment, showing improved accuracy compared with GoogLeNet [46], AlexNet [47,48,49], and Visual Geometry Group networks [50]. Huang et al. [51] upgraded MTI-100 to MTI-200a to enable crack detection in the concrete lining of metro tunnels by training the widely used fully convolutional network model for semantic segmentation of surface damage. The two-stream algorithm of the fully convolutional network model was superior to the region-growing and adaptive thresholding algorithms of existing IPMs in inference time and error rate [52,53]. Song et al. [54] proposed a deep learning-based tunnel crack detection system using the DeepLab model. However, they identified dataset scarcity and the challenges associated with labeling, which should be addressed in future research. Li et al. [55] developed the Metro Tunnel Surface Inspection System based on Faster RCNN for the high-precision automatic detection of defects in tunnel concrete linings. The system improved the location and classification of cracks, spalling, and leaks; however, it is insufficient for providing data for facility condition assessment. Moreover, the limitations in collecting sufficient datasets and high-quality images at high speeds should be addressed in the future for quantitative damage assessment.

Despite successfully detecting cracks using numerous DL technologies, several critical technical challenges remain [56]. In practice, the performance of CV-based crack damage detection models is significantly influenced by the quality of crack images collected under various conditions [57]. Mobile scanning systems are the most efficient for safely collecting tunnel images. However, vehicle vibrations and the limitation of close-up imaging of specific targets can lead to motion blur and resolution deficiencies in the collected images. These issues can result in the loss of image information, making crack detection more challenging and leaving fine cracks with widths of <0.3 mm undetected.

### 2.2. IQA for Motion Blur

With advancements in science and technology and increased availability, camera-based MTSSs are becoming prominent in tunnel maintenance. However, motion blur frequently arises when capturing images under high-speed and low-light conditions [57]. Research on image deblurring to address motion blur has focused on the iterative estimation of the blur kernel, known as the point spread function, from a blurred image—known as blind deconvolution [58,59]. However, non-blind deconvolution uses an inertial measurement unit-based deblurring approach to estimate uniform/non-uniform motion blur kernels from inertial measurement unit readings, encoding motion information into the data for later retrieval [60]. However, the combination of camera and subject movement in real images generates more complex, non-uniform blur.

Recently, deep neural networks have been used to address issues related to non-uniform blur kernels. However, deep neural networks require large training datasets to enhance the performance of trained models [60]. Several image-deblurring datasets, such as Need for Speed [61], DeBlurNet [62], GOPRO [63], Realistic and Diverse Scenes dataset [64], Human-aware Image Deblurring [65], and Real-World Blur Dataset [66], are publicly available. The blurred images in these datasets are generated by capturing video at high frame rates of 120 or 240 frames per second (FPS). One frame from the video is used as a sharp reference image. Subsequently, blurred images are synthesized sequentially around the sharp reference frame. Different levels and extents of motion blur effects are produced by varying the number of consecutive frames averaged in this manner [67].

Accurately evaluating images degraded by motion blur is crucial [60]. Image quality can be assessed subjectively (qualitatively) through human vision or objectively (quantitatively) using numerical metrics [68]. Subjective approaches to IQA are impractical for image processing applications. Objective approaches define quantitative measures that represent perceived image quality. Objective image quality metrics can be categorized into three types based on dependence on reference images: full reference IQA (FR-IQA), reduced reference IQA (RR-IQA), and no reference IQA (NR-IQA) [69].

FR-IQA measures image quality using the original, undistorted, and distorted images [70]. In FR-IQA, the Peak Signal-to-Noise Ratio (PSNR) and Structural Similarity Index Metric (SSIM) are the most widely used metrics for evaluating image deblurring algorithms. These metrics do not directly assess image quality; however, they reveal the extent of differences by comparing reconstructed and original images. PSNR is an index used to evaluate the quality of an image degraded by noise. It measures the similarity between a restored image and the original sharp image by calculating the ratio of the maximum power of the signal, represented by the mean-squared error between the two images, to the power of the signal noise. A higher PSNR indicates greater similarity to the original image [71]. SSIM evaluates blur levels (BLs) by comparing three components—luminance, contrast, and structure—between two images, with scores of 0–1, where a score closer to 1 indicates greater similarity to the original image. As a function of brightness and contrast, the SSIM is highly sensitive to parameter changes [72]. Abdullah-Al-Mamun et al. [60] proposed the BL metric to estimate motion profiles by determining the extent of degradation caused by motion, such as pixel shifts and rotations in images. The BL metric outperformed SSIM and PSNR in explaining motion blur and sharpness in low-light and low-quality images. However, it did not provide information on the direction of blurred motion in test image datasets.

The NR-IQA algorithm can provide quality assessments of images without requiring a reference image or specific features; nonetheless, it is more challenging than FR-IQA. Owing to the absence of a reference image, the modeling process must consider the statistics of the reference image, human perceptual characteristics, and the impact of distortions on image statistics. Evaluating the effectiveness of quality measurements on specific distorted images without reference images is also limited [73].

RR-IQA algorithms assess the quality of distorted images using limited features of reference images instead of complete images [74]. Various databases are used to evaluate the suitability of developed RR-IQA metrics. Publicly available databases for evaluating developed quality metrics include LIVE2005 [75], TID2008 [76], TID2013 [77], and ILV [78]. However, no image databases are tailored to tunnel environments, limiting the assessment of the quality of images acquired via MTSSs.

The image sharpness assessment of the Imatest^®^ software (version 24.1) part of the RR-IQA, requires impractical test charts for natural images or common dataset-based training. Moreover, such measurements cannot provide information on the direction of motion, causing blurred results [60]. However, Imatest^®^ analyzes camera resolution, enabling the performance analysis of cameras capturing images [79]. Obtaining quantitative motion-blurred image data in an indoor setting that simulates tunnel conditions is possible by adjusting physical movement and lighting conditions to create a low-light environment and capturing test charts. This can be achieved based on the movement speed and exposure performance of the camera. A database compiled from these images would aid in assessing and analyzing image quality. Based on the preceding literature review, addressing image data limitations for applying DL and evaluating image quality is essential. Therefore, acquiring motion-blurred image data and conducting IQAs in a standardized indoor environment are necessary.

## 3. Methodology

### 3.1. Modulation Transfer Function (MTF)

The digital images acquired via camera-based tunnel scanning systems undergo multiple processes, including image generation, compression, storage, and transmission. The final image quality will degrade if visual information is lost at any of these stages [80]. A reliable IQA metric is essential for selecting high-quality images. Human subjective image evaluations are reliable; nevertheless, their application requires considerable time and effort [81]. An objective IQA that accurately reflects the perceptual characteristics of the human visual system is necessary [82]. The MTF is a well-established method for measuring sharpness for camera systems [83].

Spatial resolution and image sharpness are fundamental characteristics of digital imaging devices such as MTSSs. The international standard ISO 12233 for digital cameras provides guidelines and evaluation methods for determining image sharpness and resolution [84]. The MTF and spatial frequency response (SFR) of optical imaging systems are metrics defined by ISO 12233 for spatial resolution analysis. Figure 1 presents the standardized Imatest ISO 12233:2017 edge spatial frequency response (eSFR) target chart for measuring image quality parameters. This chart fully complies with the low-contrast eSFR—ISO standard—and integrates several image quality parameters, including sharpness, lateral chromatic aberration, white balance, tone response, color accuracy, and noise [85].

The slanted-edge method of measuring SFR, as presented in the international standard ISO 12233, is a technique for deriving the MTF from a slanted edge on a standardized test chart [86]. Sharpness is an important image quality factor, defined as the boundary between areas of different tones. It is quantified by the edge width in the image, which can be determined by measuring the pixel-level distance between 10% and 90% of the final value. This edge width is assessed in the frequency domain, where the frequency is measured in cycles per distance or line pairs (millimeters, inches, pixels, image height, or sometimes angle [degrees or milliradians]) [87]. The relative contrast at a given spatial frequency (output/input contrast) is referred to as MTF. In Imatest^®^, MTF is used interchangeably with SFR.

Figure 2 is a scheme of the slanted-edge method. The tilted edge of the eSFR ISO test chart has an angle of 5°. The region of interest (ROI) is depicted as a rectangular area along the short side (Figure 2a). Figure 2b presents a one-dimensional edge spread function (ESF) graph, approximated using a finite difference filter. The discrete Fourier transform, calculated with a Hamming window, is displayed. Figure 2c presents the MTF, represented as the normalized value of the complex coefficients of the discrete Fourier transform. The MTF is the Fourier transform of the impulse response, which is the derivative of the edge response. The MTF of a sampled image is a measure of image resolution and sharpness. Thus, it is used to determine the level of detail a camera can reproduce [88].

MTF50 represents the spatial frequency at which the contrast falls to half (50%) of its low-frequency value, while MTF50P indicates the spatial frequency at which the contrast drops to half of its peak value [89]. Lower MTF values signify poorer image quality. Koren [90] confirmed that MTF50 values measured using the Imatest^®^ software were closely correlated with perceived sharpness by the human eye.

Figure 3 demonstrates the sharpness of an edge using the Imatest^®^ software to derive rise distance and MTF [91]. The blurred edge width (BEW) is the width of the rise distance from 10% to 90% measured from the ESF, expressed in pixels. The BEW of the reference image is 1.58 pixels, whereas that of the image affected by motion blur increases to 2.63 pixels. The slope width of the rise distance graph increases as motion blur occurs, indicating reduced image sharpness. This analytical method quantitatively assesses image sharpness.

### 3.2. High-Speed Translational Moving Panel Device

In the review of existing literature, motion-blurred image datasets available to the public for IQA are regarded as virtual data. The motion blur in images of moving objects is influenced more by the camera’s exposure than by FPS. During the camera’s shutter exposure time, a moving object is captured by integrating all positions along its trajectory, resulting in a blurred image. Dinh et al. [83] developed an indoor rotational test device with a maximum of 300 RPM to evaluate the quality of motion-blurred images. In the experiments, the camera remained stationary while the chart rotated. The slope of the ESF gradually increased with increasing exposure time owing to differences in lighting brightness and rotational speed, indicating motion blur. However, MTF evaluation based on the slanted-edge method is more suitable for translational than rotational motion. Luo et al. [92] conducted an experiment where a smartphone was moved in a translational direction at 1 m/s while recording a fixed chart. Motion blur caused the ESF to widen and the MTF profile to decrease. Their analysis indicated that higher FPS reduced motion blur, and exposure speed was expected to impact high-speed imaging. Nevertheless, no study has investigated the effects of motion blur in images captured at high speeds of 70 km/h (~19.4 m/s).

We designed a novel device that captures images while moving at high speeds in a translational direction (Figure 4). The device can achieve a maximum speed of 110 km/h, with adjustments of 10 km/h increments. In the developed high-speed translational moving panel device, image capture was initiated at a speed of 50 km/h when the motor power converged to approximately 25%, maintaining a steady state as shown in Figure 5. Notably, once the motor power fluctuation remained consistent within ±0.1%, the speed also remained constant.

The panel can use the Imatest^®^ eSFR ISO test chart for MTF measurement. The setup was constructed according to the shooting method outlined in ISO 12233 (Figure 6a). Two 120 W daylight LED floodlights with a color temperature of 5600 K were positioned at a 45° angle to the front of the test chart (Figure 6b). The machine vision area scan camera has a complementary metal-oxide–semiconductor image sensor with a 4096 × 2304 resolution, a minimum exposure speed of 5 μs, and a global shutter. The shooting distance was 2 m, with an image resolution of 0.2 mm/pixel. The rotational test device will be used to determine factors for evaluating the quality of images captured by high-speed moving tunnel scanning systems. This device can serve as a standard imaging setup for evaluating the quality of images captured by high-speed MTSS in motion.

### 3.3. Indoor Test Setup in Standard Environments Considering Camera Exposure

In low-light environments such as tunnels, short exposure times of image sensors can introduce noise, resulting in coarse images. Conversely, longer exposure times can reduce noise, but are highly susceptible to motion blur [93]. A solution to this limitation is using high-intensity lighting to supply sufficient brightness to the image sensor, even at short exposure times. The exposure performance of a camera is influenced by the shutter speed, ISO, and lens aperture (F-stop) values. Finding the optimal exposure settings that make moving objects appear stationary is crucial for capturing sharp images. Sasama et al. [17] proposed that, in a railway scanning system, to acquire images at a resolution of 1 mm/pixel while moving at a speed of 20 km/h, an illuminance of 20,000 lx on the surface is required. Therefore, to meet the fast exposure performance of the camera, the lighting brightness was set to 15,000 lx and 40,000 lx. At illuminance levels below 15,000 lx, images captured at a shutter speed of 50 µs were not identifiable.

Videos captured by the high-speed translational moving panel device contain black background areas and the test chart. From the video frames capturing the entire test chart, 30 test chart images were extracted for each case (Figure 7) to construct a dataset of high-speed motion blur images.

An indoor test was conducted (Table 1) considering the panel’s movement speed and the camera’s exposure performance. The aperture value (F) of the lens was 2.8, and the FPS was 100. The shutter speed varied at 50 μs, 100 μs, 250 μs, and 500 μs, while the ISO was adjusted to 640, 1250, and 1600. The illuminance on the chart surface was 15,000 lx and 40,000 lx. The speed of the high-speed translational moving panel was varied at 0 km/h (stationary image), 10 km/h, 30 km/h, 50 km/h, and 70 km/h.

## 4. Results

An IQA was performed on images captured from indoor experiments based on the international standard ISO 12233. We employed existing IQA metrics, including the ESF of the RR-IQA for BEW and MTF50 and the PSNR and SSIM for FR-IQA. The metrics were analyzed according to variations in illuminance, shutter speed, and movement speed.

### 4.1. Analysis of BEW and MTF50 by Moving Speed and Shutter Speed

Figure 8 illustrates the test chart captured at an illuminance of 15,000 lx, with variations in the speed of the moving panel and the camera’s shutter speed. According to subjective visual analysis, motion blur increases as the moving panel speed increases. Conversely, the motion blur decreases as the camera’s shutter speed increases. Despite increasing the ISO sensitivity, the images appeared darker as the shutter speed increased. Figure 9 illustrates the ROI of the slanted edge in images captured at 70 km/h, with varying shutter speeds. The motion blur decreased as the shutter speed increased from 500 μs to 50 μs. However, the level of light detected by the image sensor decreased owing to the higher shutter speed, resulting in darker images. Nonetheless, changes in the moving speed did not affect the image brightness for the same shutter speed.

The Imatest^®^ software was used to objectively assess motion blur. Table 2 presents the BEW and MTF50 of the ESF for the central position of the eSFR ISO test chart. These values are shown in Figure 10. Table 2 presents the BEW and MTF50 of the ESF for the central position of the eSFR ISO test chart. As the speed of the moving panel increased, the BEW of the ESF increased, and the MTF50 decreased. This indicates that motion blur in the images increased with increasing speed. At 70 km/h, as the camera’s shutter speed increased from 500 μs to 50 μs, the BEW decreased from 38.28 pixels to 5.32 pixels. Simultaneously, the MTF50 increased from 0.0132 cycles/pixel to 0.0981 cycles/pixel, indicating reduced motion blur. The BEW and MTF50 metrics correspond to the visual observations of motion-blurred images.

In the experiment with 15,000 lx illuminance, higher shutter speeds resulted in darker images. Images captured at 50 μs appeared darker to the naked eye than those captured at 500 μs. Contrast is a crucial factor in image quality. The experiment was repeated with an illuminance of 40,000 lx to assess the impact of illuminance on motion blur. Figure 11 presents the test chart captured under 40,000 lx illuminance with varying moving and camera shutter speeds. Table 3 presents the average and standard deviation of BEW and MTF50 for the ESF at the central position of 30 eSFR ISO test charts. These values are shown in Figure 12. Similar to the 15,000 lx experiment, the BEW increased and MTF50 decreased as the moving speed increased, indicating an increase in motion blur.

Through graphical analysis, it was observed that both moving and shutter speed considerably influenced BEW and MTF50. To statistically validate this observation, a two-way analysis of variance (ANOVA) was conducted, and the results are presented in Table 4. The analysis was conducted separately by illuminance condition, and the effects of moving speed, shutter speed, and their interaction on the dependent variables BEW and MTF50 were evaluated.

According to the two-way ANOVA results, under the 15,000 lx illuminance condition, shutter speed had the greatest effect on BEW (F = 43,840.35, *p* < 0.0001), while travel speed (F = 29,998.22, *p* < 0.0001) also showed a highly significant effect. The interaction term between these two variables (F = 8298.73, *p* < 0.0001) was significant, indicating that BEW was not influenced by a single factor, but rather by the combined effects of speed and shutter speed.

Under the same condition, MTF50 was most influenced by shutter speed (F = 233.39, *p* < 0.0001), while both speed (F = 365.26, *p* < 0.00001) and the interaction term (F = 16.48, *p* < 0.00001) also showed statistically significant effects. This suggests that image sharpness also changed nonlinearly owing to the interaction between the two variables.

The overall trend remained consistent under the higher illuminance condition of 40,000 lx. For BEW, both shutter speed (F = 51,130.25, *p* < 0.0001) and speed (F = 36,821.60, *p* < 0.0001) showed strong significance and explanatory power. Regarding MTF50, shutter speed (F = 523.77, *p* < 0.0001) had the largest effect, while speed (F = 1265.27) and the interaction term (F = 59.73) were also significant.

Notably, the F-values for shutter speed were the highest under both lighting conditions, with increases of approximately 16.6% for BEW and 124.4% for MTF50. This indicates that the influence of shutter speed remained consistent, even in high-illuminance environments, and that in some cases, its relative explanatory power may have even increased. However, this does not necessarily imply an absolute increase in effect size; rather, it suggests that the shutter speed control remains significant, even with increased illuminance, and may be more effective under certain conditions.

### 4.2. Analysis of Image Quality Variation Due to Increased Illuminance

To quantitatively analyze the impact of illuminance changes on image quality, heatmaps were generated to visualize the variation rates of BEW and MTF50 when increasing illuminance from 15,000 lx to 40,000 lx, across varying travel speeds (0–70 km/h) and shutter speeds (50–500 μs). The variation rate, calculated as a percentage using Equation (1), compares the relative increase or decrease in values at 40,000 lx to those under 15,000 lx, serving as the reference baseline.(1)$$Change\;Rate\left(\%\right)=100\times \frac{40,000\;lx-15,000\;lx}{15,000\;lx}$$

Figure 13 shows a heatmap of the BEW change rate, where negative values (blue) indicate that blur decreased with increased illuminance, signifying improved image quality. Conversely, positive values (red) indicate increased blur, meaning image quality degradation. The analysis indicated that, under most high-speed conditions (≥50 km/h), BEW decreased by more than 5% with increased illuminance, suggesting that high illuminance positively influences blur suppression. Notably, under the condition of a shutter speed of 250 μs at 70 km/h, BEW decreased by approximately 8.1%. However, under low-speed conditions (0–10 km/h), BEW either showed minimal change or increased, indicating that higher illuminance has limited impact on blur at low speed.

Figure 14 shows a heatmap of the MTF50 change rate, where negative values (blue) indicate a decrease in sharpness owing to increased blur, while positive values (red) represent improved sharpness. The MTF50 results exhibited more complex patterns. Under certain conditions (e.g., shutter speeds of 100 and 250 μs and speed of 0 km/h), MTF50 increased by 5–10% or more, confirming improved image sharpness. However, under high-speed conditions—particularly a shutter speed of 500 μs at 70 km/h—sharpness decreased by 31.5%, even with increased illuminance. This suggests that, beyond a certain level, higher illuminance may cause sensor overexposure or increased specular reflection, thereby reducing sharpness.

These heatmap analyses demonstrated that image quality varies non-linearly based on the interaction between illuminance and shutter speed combinations. The findings quantitatively confirmed that increasing illuminance does not always enhance quality. Therefore, when operating or developing an MTSS, illumination design should be optimized for shooting conditions. Moreover, when using MTF50 as a primary quality metric, a careful balance between illuminance and shutter speed is required.

After visualizing the effects of illuminance changes on BEW and MTF50 across conditions using heatmaps, independent-sample *t*-tests were performed to assess the statistical significance of these changes. Table 5 presents the resulting *p*-values for each condition.

Concerning BEW, 10 out of 20 experimental combinations (50%) exhibited statistically significant differences at the *p* < 0.05 level. Notably, repeated significance was found under high-speed conditions (≥30 km/h) and long shutter durations (≥250 μs). For example, conditions such as (30 km/h, 250 μs), (50 km/h, 500 μs), and (70 km/h, 250 μs) yielded a *p*-value of 0, indicating highly significant differences. Even under mid-speed conditions (10–50 km/h), statistical significance was found when the shutter speed was 250 μs or 500 μs (*p* = 0.0030–0.0072). However, under 0 km/h or short shutter speed conditions (50–100 μs), most cases did not exhibit statistical significance. These results suggest that illuminance change alone does not consistently affect BEW; however, under specific conditions (high speed + long exposure), it can statistically induce either suppression or worsening of motion blur.

Conversely, for MTF50, only 3 out of 20 conditions (15%) showed statistically significant differences at the *p* < 0.05 level. All of these significant results occurred under stationary conditions (0 km/h) at shutter speeds of 100 μs (*p* = 0.0014), 250 μs (*p* = 0.0003), and 500 μs (*p* = 0.0435). However, under moving conditions of 10 km/h or more, none of the shutter combinations showed statistical significance. For instance, under 10 km/h (250 μs), *p* = 0.3344, and under 50 km/h (250 μs), *p* = 0.5823, indicating that illuminance change had no statistically significant effect on image sharpness. These findings suggest that MTF50 was sensitive to illuminance changes only under certain conditions, primarily in stationary environments. Under more realistic operating conditions involving motion, the influence was minimal. Specifically, all statistically significant conditions occurred under stationary conditions with long exposures (≥100 μs), suggesting that illuminance changes may have affected optical factors, such as camera exposure compensation and illuminance uniformity.

Ultimately, the influence of illuminance change on MTF50 was statistically significant only under limited conditions, and thus cannot be generalized based on the results of this study. This implies that, rather than simply increasing illumination, a comprehensive design that quantitatively considers the interaction between speed, shutter, and illuminance is essential for optimizing image quality.

To supplement the finding that BEW and MTF50 showed no significant differences under many conditions in the statistical significance test, boxplots were generated for each metric to visually compare their distributions (Figure 15). This analysis used 600 measured BEW and MTF50 values obtained under 15,000 lx and 40,000 lx conditions. As presented in Table 6, each boxplot is based on the median, first quartile (Q1), third quartile (Q3), and interquartile range (IQR = Q3 − Q1). Outliers are defined as values smaller than Q1 − 1.5 × IQR or greater than Q3 + 1.5 × IQR, and are shown as individual points.

The boxplot in Figure 15a shows BEW measurements from 1200 motion blur images, with values ranging from approximately 2.6 to 39.1 pixels. Across both illuminance conditions (15,000 lx and 40,000 lx), the means, medians, and interquartile ranges were similar. Specifically, the median BEW was 5.13 px under 15,000 lx and 5.10 px under 40,000 lx, showing virtually no difference. The IQRs were 3.94–10.17 px and 3.79–9.59 px, respectively, and the mean values were 9.48 px at 15,000 lx and 8.95 px at 40,000 lx. This aligns with the statistical significance test result (*p* = 0.291), indicating no significant effect. These findings suggest that increasing illuminance has minimal impact on BEW, and it is difficult to expect blur improvement solely through changes in lighting conditions.

However, large numbers of outliers were observed in the BEW distribution, primarily at a speed of 60–70 km/h combined with 250–500 μs shutter speeds. In these cases, BEW values sharply increased to over 30 pixels, indicating significant motion blur when high speed coincides with long exposure. The locations and number of outliers were similar across both lighting conditions, suggesting that illumination change contributes little to suppressing extreme blur artifacts.

In Figure 15b, the MTF50 values ranged between approximately 0.01 and 0.20 cycles/pixel. Similar to BEW, the medians and IQRs across both 15,000 lx and 40,000 lx conditions were highly identical. The median MTF50 values were 0.1016 (15,000 lx) and 0.1005 (40,000 lx), while the IQRs were approximately 0.053–0.141. This visually confirms that increased illuminance did not significantly enhance overall image sharpness. The mean MTF50 values were 0.0960 (15,000 lx) and 0.0961 (40,000 lx), consistent with the previously conducted statistical significance test (*p* = 0.998).

From the boxplot analysis, the distributions of both BEW and MTF50 under different illuminance conditions were found to be highly similar. No substantial differences were observed in medians, interquartile ranges, or overall value ranges. Although a few extreme blur values appeared as outliers in the BEW boxplot—specifically under high-speed or long-exposure conditions—the overall distribution patterns were consistent, regardless of lighting conditions.

### 4.3. Field Validation

The results of the indoor experiment confirmed that MTF is an effective metric for quantitatively evaluating motion blur caused by physical movement. However, in actual tunnel environments, where the MTSS is operated, various factors such as uneven road surfaces and vertical vibrations during driving may generate complex motion blur in multiple directions. Because the indoor testing equipment cannot fully replicate such multidirectional blur, we conducted a field experiment using the MTSS in a real tunnel to validate the applicability of MTF.

The field testbed was Songhyeon Tunnel in Incheon, South Korea, a 400 m long, three-lane one-way tunnel (Figure 16). For this validation, we used MTSS equipment fitted with a 4 K (4096 × 2) line scan camera, as shown in Figure 17 [94]. To evaluate image quality, two SFRreg test charts from Imatest [95] were attached to the tunnel’s concrete lining (Figure 18). Figure 19 presents the images captured at 20 km/h, 40 km/h, 60 km/h, and 80 km/h, with a resolution of 1 mm/pixel. The illuminance measured from a distance of 3 m during shooting was approximately 15,000 lx, and the camera exposure was set to 50 kHz, with two sets of images captured per condition.

The images in Figure 19 were taken at different vehicle speeds using the MTSS. However, visual inspection alone does not distinguish quality differences by speed. The PSNR and SSIM metrics analyzed in previous sections are FR-IQA methods, which require a static reference image for comparison against motion-blurred images. However, because the MTSS is a line scan camera-based system that inherently captures images in motion, obtaining a static reference image is not feasible. This highlights a fundamental limitation of applying FR-IQA techniques to MTSS images captured in actual tunnel environments.

Therefore, this study quantitatively analyzed the effect of motion blur on image quality in tunnel inspection settings by measuring horizontal and vertical BEW and MTF50 values at four different speeds: 20, 40, 60, and 80 km/h. Two SFRreg test charts affixed to the tunnel wall were used for the analysis. As shown in Figure 20, two ROIs were designated on each chart—one for horizontal and one for vertical MTF measurements. Eight samples were obtained for each speed condition and used for analysis.

As shown in Table 7 and Figure 21, the horizontal BEW gradually increased with vehicle speed, from 2.30 pixels at 20 km/h to 3.38 pixels at 80 km/h. In contrast, the horizontal MTF50 decreased from 0.228 cycles/pixel to 0.168 cycles/pixel, indicating a degradation in image sharpness owing to motion blur.

In the vertical direction, BEW values were consistently lower and MTF50 values were higher than those in the horizontal direction across all speed conditions. For instance, at 80 km/h, the vertical BEW was 2.19 pixels, approximately 35% lower than the horizontal value (3.38 pixels), while the vertical MTF50 was 0.234 cycles/pixel, significantly higher than the horizontal value (0.168 cycles/pixel). This directional difference suggests that motion blur primarily occurs in the horizontal direction, corresponding to the direction of travel.

Additionally, the standard deviation of the horizontal BEW increased with speed, peaking at ±1.02 pixels at 80 km/h. This implies that, at higher speeds, factors such as mechanical vibrations, exposure instability, or surface irregularities on the tunnel wall may reduce consistency in image quality. These findings validate BEW and MTF50 as effective indicators for quantitatively assessing motion blur under high-speed conditions and show their potential for further characterizing the spatial properties of blur in tunnel imagery.

## 5. Discussion

This study quantitatively analyzed the impact of motion blur—caused by high-speed driving in an MTSS—on image quality and validated the effectiveness of two evaluation metrics: MTF50 and BEW. To verify both the theoretical soundness and practical applicability of the proposed quality assessment framework, experiments were conducted in an indoor setting designed according to the ISO 12233 international standard and complemented by field tests in an actual tunnel environment.

In the indoor experiments, BEW increased and MTF50 decreased with speed, corresponding to a visually perceptible increase in motion blur. Although faster shutter speeds helped to reduce blur to some extent, they also led to underexposed and darker images due to reduced exposure time. To mitigate this, illumination was increased to 40,000 lx; however, in some cases, excessive lighting degraded sharpness across the image, suggesting that simply increasing illumination does not always improve image quality. Instead, an optimal balance between illumination and shutter speed is required.

Two-way ANOVA analysis confirmed that both BEW and MTF50 were significantly affected by vehicle speed, shutter speed, and their interaction, indicating that image quality is determined not by any single factor, but by their combined effect. BEW was particularly sensitive to long exposure times at high speeds, whereas MTF50 showed significant changes only under stationary conditions.

In the field validation at 80 km/h, the horizontal BEW measured 3.38 px, compared with 2.19 px in the vertical direction, and the horizontal MTF50 was also lower than that in the vertical direction. These results quantitatively demonstrate that image degradation is more pronounced along the horizontal axis—the direction of travel in MTSS. Additionally, the increasing standard deviation of horizontal BEW at higher speeds suggests that mechanical vibrations, road surface conditions, and lighting imbalance may compromise consistency in image quality. These findings highlight the significance of BEW and MTF50 as effective metrics for quantifying motion blur and characterizing its directional behavior in tunnel imagery.

A limitation of this study is the inability of the indoor test setup to fully simulate vertical vibration or multi-directional blur. However, because most experiments were conducted with shutter speeds faster than 50 μs, it is likely that vertical shaking had minimal effect on image quality during actual MTSS operation. This assumption was supported by the field results, where the vertical axis consistently retained higher sharpness than the horizontal. Thus, while vertical blur was not fully replicated, the focus on horizontal blur was sufficiently validated.

Moreover, recent studies have repeatedly shown that image quality significantly affects CNN-based crack detection. Models such as U-Net and Faster R-CNN are highly sensitive to input sharpness, contrast, and noise levels. Blurred or poorly lit images can result in missed detections or false positives, as cracks may not be distinguishable from the background.

To address this, attention has recently shifted toward NR-IQA methods that evaluate quality without a reference image. Metrics such as the blind/referenceless image spatial quality evaluator (BRISQUE) [96], naturalness image quality evaluator (NIQE) [97], perception-based image quality evaluator (PIQE) [98], and cumulative probability of blur detection (CPBD) [99] can quantify blur, noise, and contrast degradation. Numerous studies have reported that filtering out low-quality images using these metrics can enhance both model accuracy and computational efficiency. Specifically, datasets filtered by BRISQUE showed enhancements not only in accuracy and F1 score, but also in training efficiency [100].

Such IQA-based approaches are evolving beyond simple post-processing into preemptive quality control strategies that enhance dataset reliability and CNN training efficiency. In high-speed environments such as MTSS, attaching reference targets such as ISO 12233 charts is impractical, and FR-IQA methods such as PSNR or SSIM are infeasible owing to the lack of ground-truth images. Thus, NR-IQA offers a practical and scalable solution for real-time quality assessment and integration into automated tunnel inspection systems.

Future research should focus on applying various NR-IQA metrics to real tunnel environments and quantifying how CNN crack detection performance varies with image sets filtered based on those metrics. This could lead to an integrated framework linking IQA, dataset management, and model performance improvement.

## 6. Conclusions

This study quantitatively analyzed the impact of motion blur generated during high-speed driving on the image quality of an MTSS and validated the applicability of two objective evaluation metrics: MTF50 and BEW. Using an ISO 12233-based indoor experimental setup along with data collected in an actual tunnel environment, this study demonstrated that these metrics serve as practical tools for effectively assessing image degradation during high-speed movement.

The experimental results revealed that, as driving speed increased, horizontal BEW increased while MTF50 decreased, quantitatively confirming that horizontal motion blur is the primary factor contributing to image degradation owing to the directional characteristics of MTSS travel. Conversely, the vertical direction exhibited lower BEW and higher MTF50 values, indicating that vertical blur has a relatively minor effect. The effects of illumination and shutter speed combinations on image quality also varied—excessive lighting sometimes reduced image sharpness. This finding emphasizes that higher illumination alone does not guarantee improved quality and highlights the need for an optimized balance between illumination and shutter speed.

Additionally, BEW and MTF50 have been demonstrated as reliable metrics that quantitatively represent blur extent and sharpness, respectively, and are applicable for IQA, even under complex operating conditions. Notably, this study provides practical contributions by exploring the directional characteristics of motion blur through horizontal–vertical comparisons using BEW and MTF50 in real tunnel environments, a gap previously unaddressed in the literature.

These findings can be utilized to optimize camera settings in MTSS, establish image quality standards based on driving conditions, and develop real-time image quality monitoring systems. They offer a technical foundation for securing image quality in high-speed data acquisition environments.

Future research should experimentally simulate more sophisticated multi-directional motion blur conditions, such as vertical vibration and rotational blur. It should also incorporate NR-IQA methods to enable automated image filtering and integrate with CNN-based crack detection, establishing a comprehensive intelligent image quality management framework. Such advancements will enhance the reliability and precision of automated tunnel inspection systems operating under high-speed conditions.

## Figures and Tables

**Figure 1 sensors-25-03804-f001:**
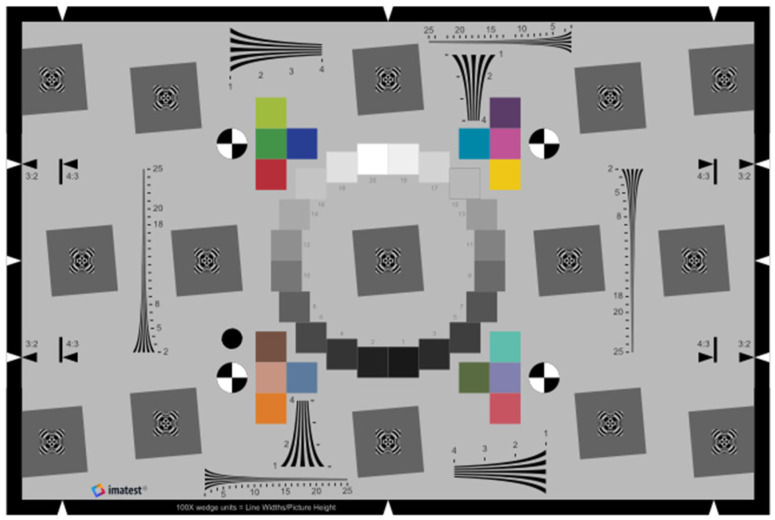
Enhanced eSFR ISO test chart: 3:2 aspect ratio, 6 added squares on sides, 16 added color patches, and several added wedge patterns.

**Figure 2 sensors-25-03804-f002:**
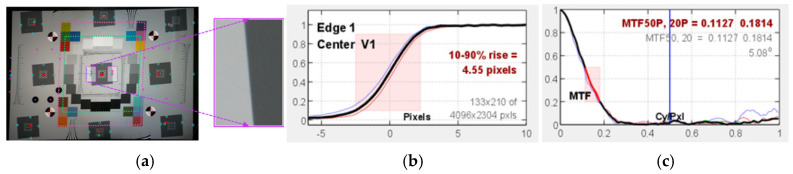
Illustration of the slanted edge-based modulation transfer function (MTF) estimation process: (**a**) selection of a region of interest (ROI), (**b**) normalized edge spread function, and (**c**) estimated MTF.

**Figure 3 sensors-25-03804-f003:**
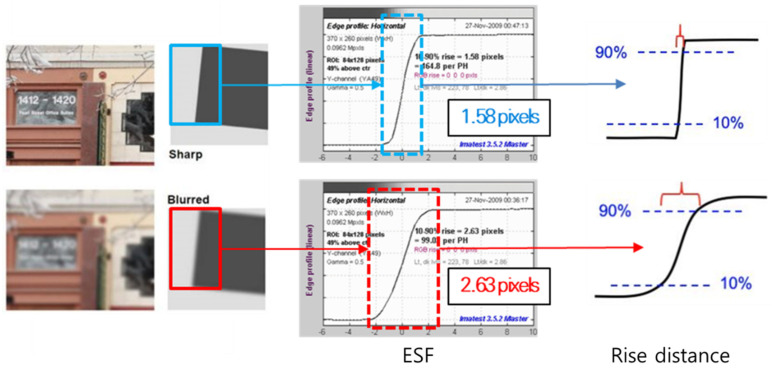
Illustration of the 10–90% rise distance graph on blurry and sharp edges.

**Figure 4 sensors-25-03804-f004:**
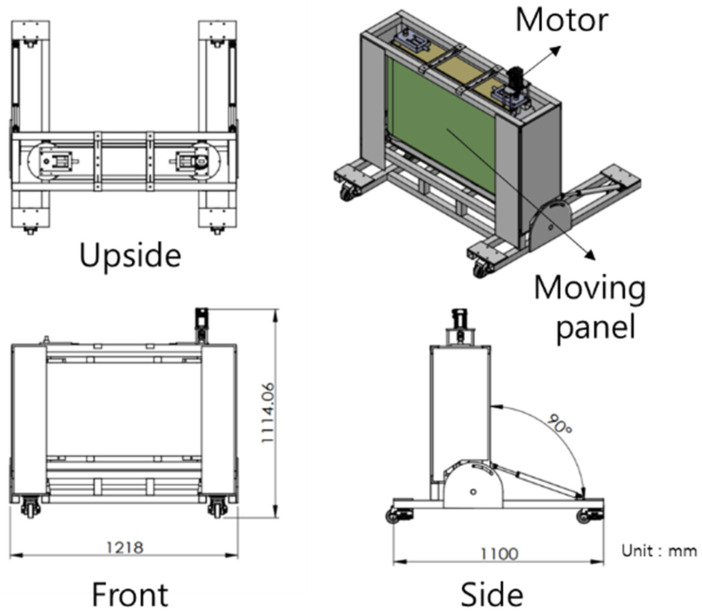
Illustration of the high-speed translational moving panel device.

**Figure 5 sensors-25-03804-f005:**
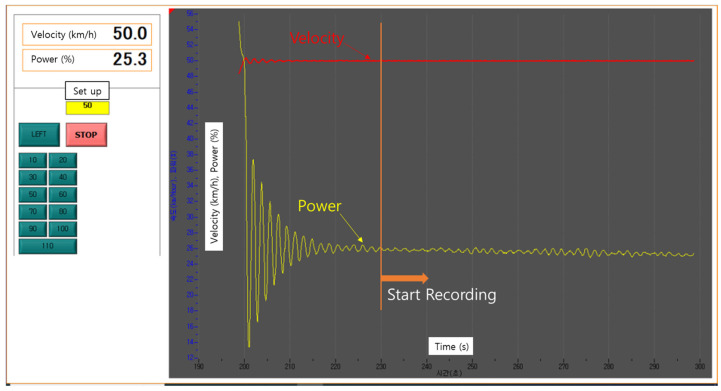
Velocity control of high-speed translational moving panel device.

**Figure 6 sensors-25-03804-f006:**
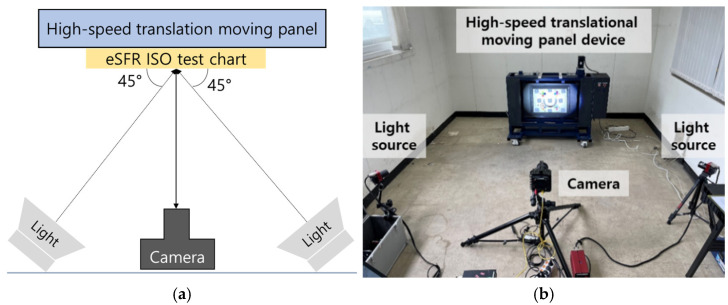
Motion blur capturing on a high-speed translational moving panel in a laboratory: (**a**) Schematic of the motion blur capturing setup and (**b**) eSFR ISO test chart on the panel.

**Figure 7 sensors-25-03804-f007:**
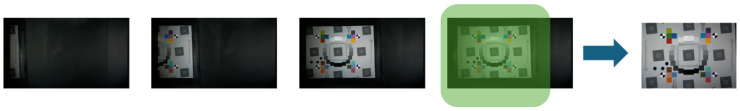
Motion-blurred images captured from a high-speed translational moving panel device.

**Figure 8 sensors-25-03804-f008:**
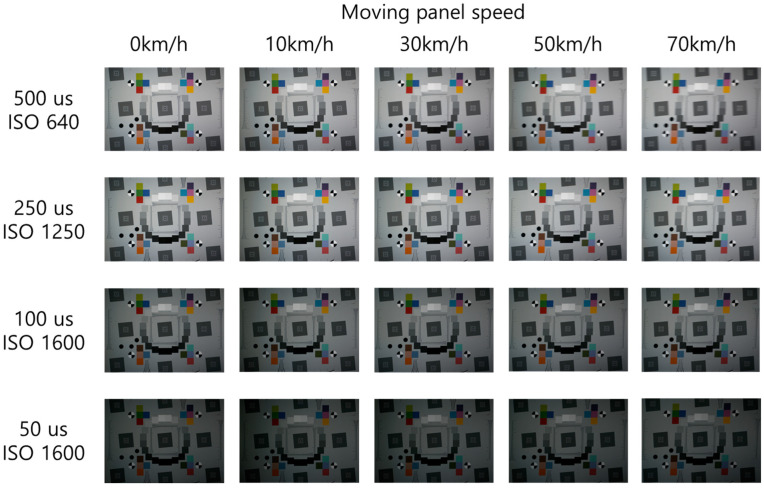
Images captured using the moving panel device for motion blur at 15,000 lx.

**Figure 9 sensors-25-03804-f009:**
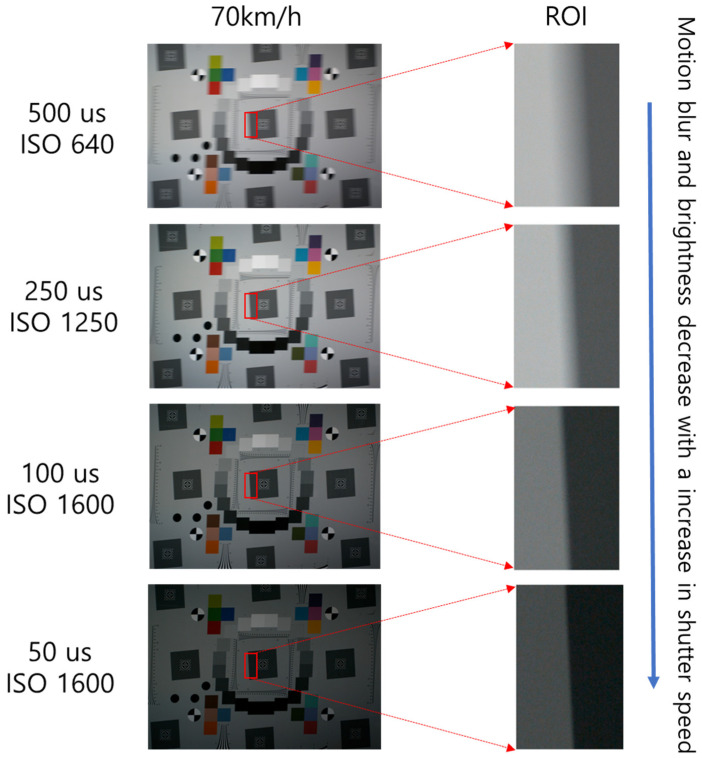
Comparison of motion blur as shutter speed changes at 15,000 lx.

**Figure 10 sensors-25-03804-f010:**
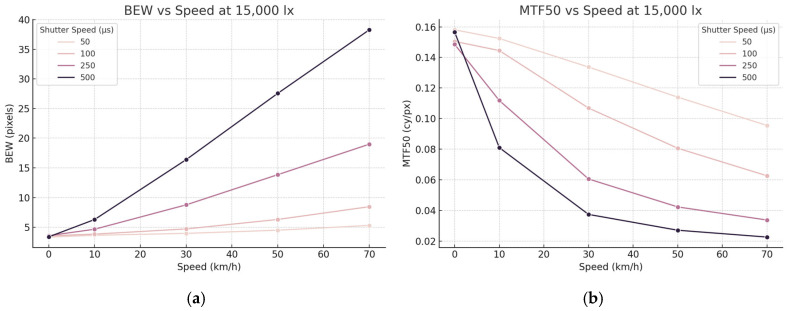
Image quality trends at 15,000 lx illuminance. (**a**) BEW increases with speed, particularly under longer shutter durations. (**b**) MTF50 decreases as speed increases, with sharper degradation observed at slower shutter speeds.

**Figure 11 sensors-25-03804-f011:**
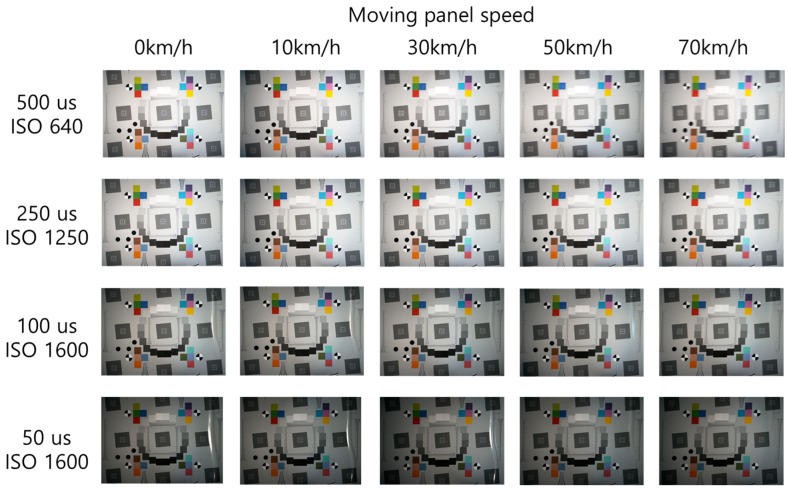
Images captured using the moving panel device for motion blur at 40,000 lx.

**Figure 12 sensors-25-03804-f012:**
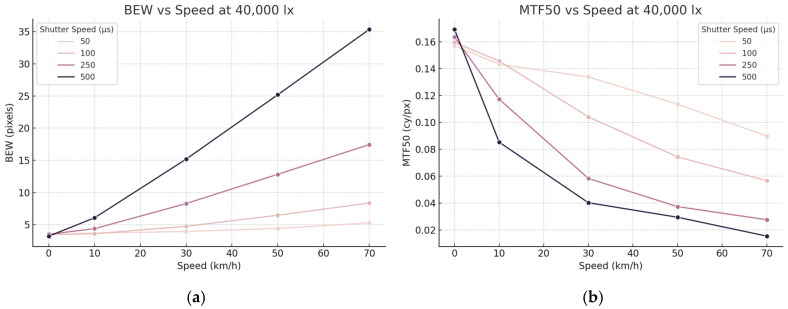
Image quality trends at 40,000 lx illuminance. (**a**) BEW increases with speed, particularly under longer shutter durations. (**b**) MTF50 decreases as speed increases, with sharper degradation observed at slower shutter speeds.

**Figure 13 sensors-25-03804-f013:**
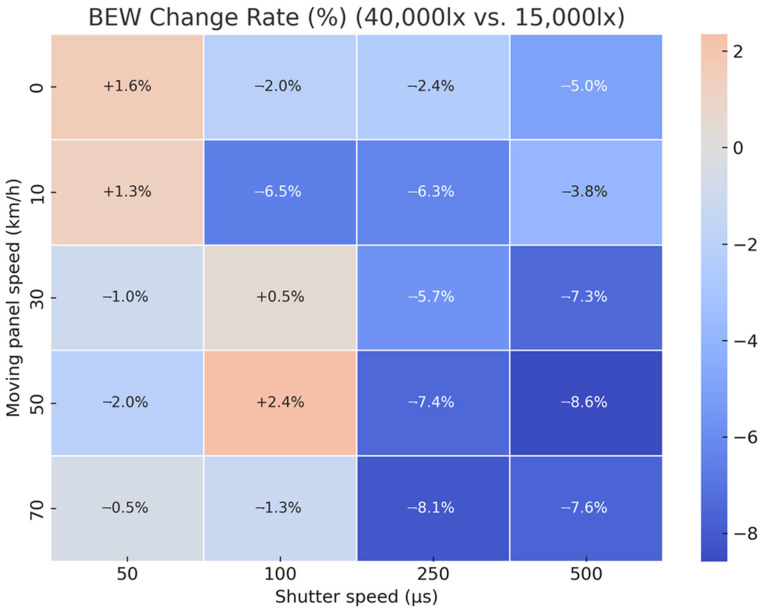
Heatmap of BEW change rates (%) owing to illuminance increase from 15,000 lx to 40,000 lx across varying speeds and shutter durations.

**Figure 14 sensors-25-03804-f014:**
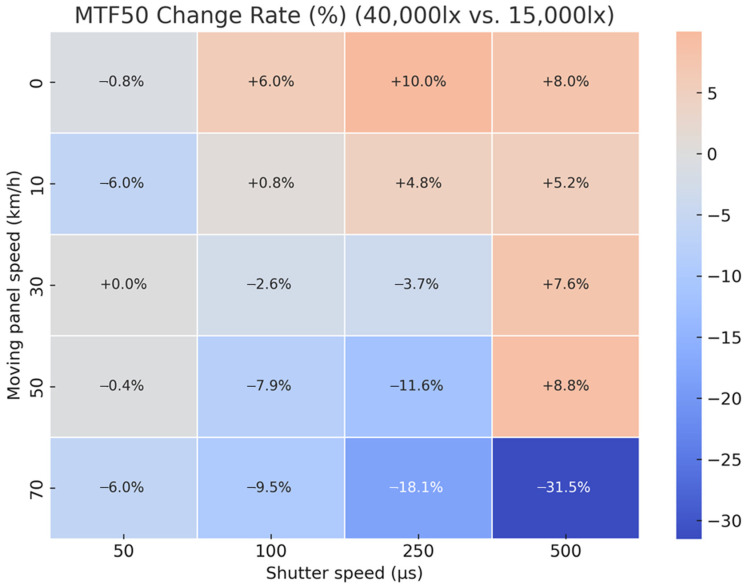
Heatmap of MTF50 change rates (%) resulting from illuminance variation (15,000 lx → 40,000 lx) under different speed and exposure settings.

**Figure 15 sensors-25-03804-f015:**
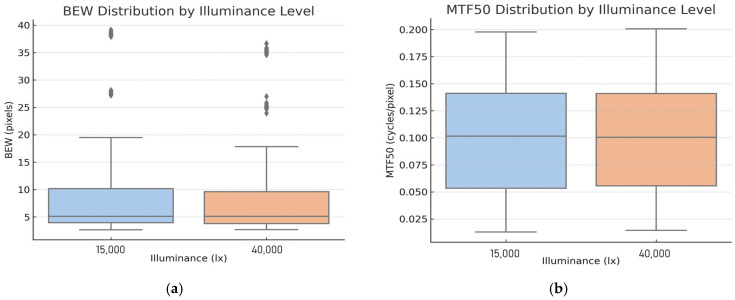
Boxplots showing the distribution of image quality metrics under two illuminance conditions (15,000 lx and 40,000 lx). (**a**) BEW indicates the extent of motion blur, and (**b**) MTF50 represents image sharpness.

**Figure 16 sensors-25-03804-f016:**
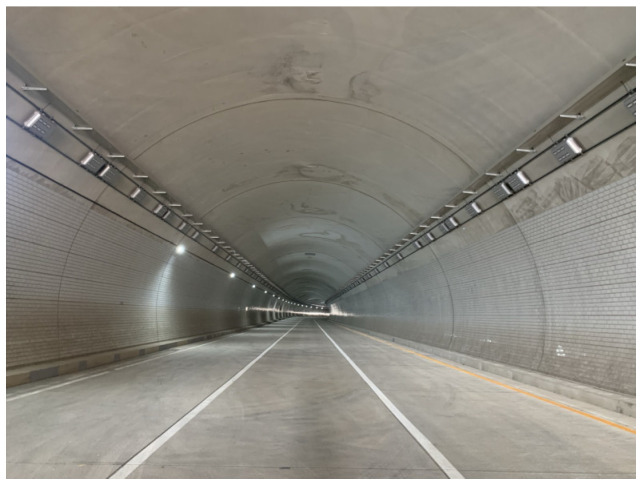
View of the Songhyeon Tunnel testbed used for field validation.

**Figure 17 sensors-25-03804-f017:**
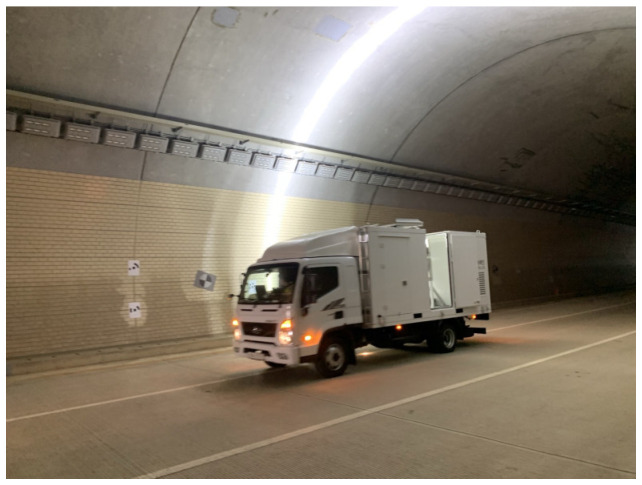
MTSS equipped with 4 K (4096 × 2) line-scan cameras used for image acquisition during field testing in the Songhyeon Tunnel.

**Figure 18 sensors-25-03804-f018:**
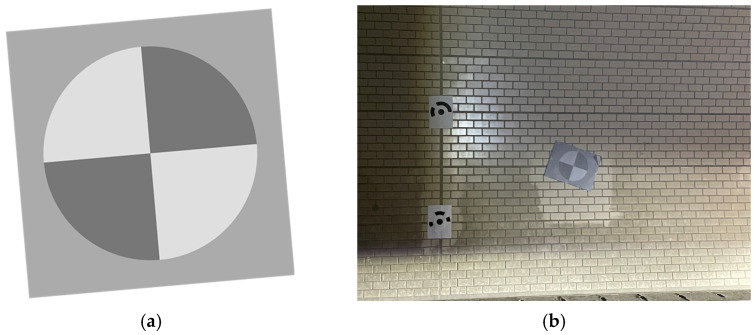
SFRreg test charts used for evaluating motion blur and spatial resolution in tunnel environments; (**a**) SFRreg test chart, (**b**) installation of SFRreg test charts on the tunnel lining wall.

**Figure 19 sensors-25-03804-f019:**
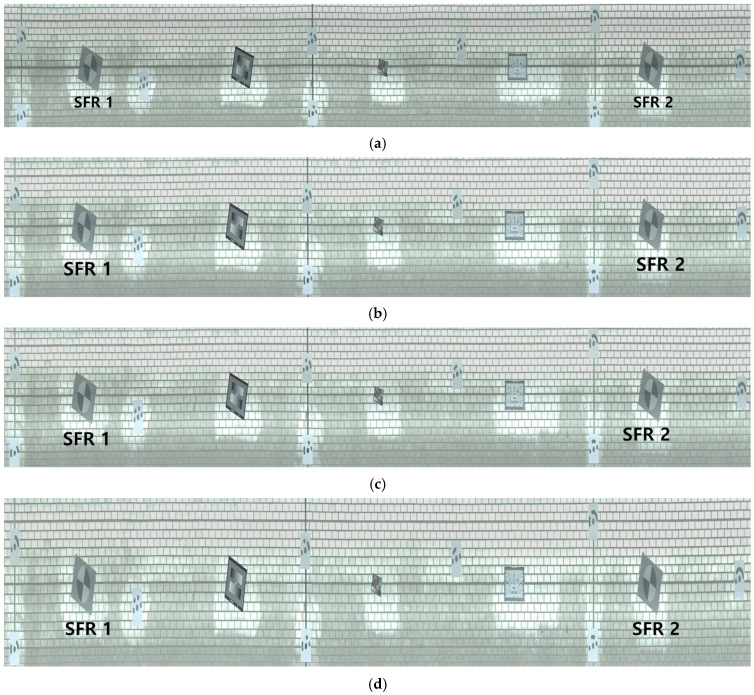
SFRreg test chart images acquired using the MTSS at various vehicle speeds. (**a**) 20 km/h, (**b**) 40 km/h, (**c**) 60 km/h, and (**d**) 80 km/h.

**Figure 20 sensors-25-03804-f020:**
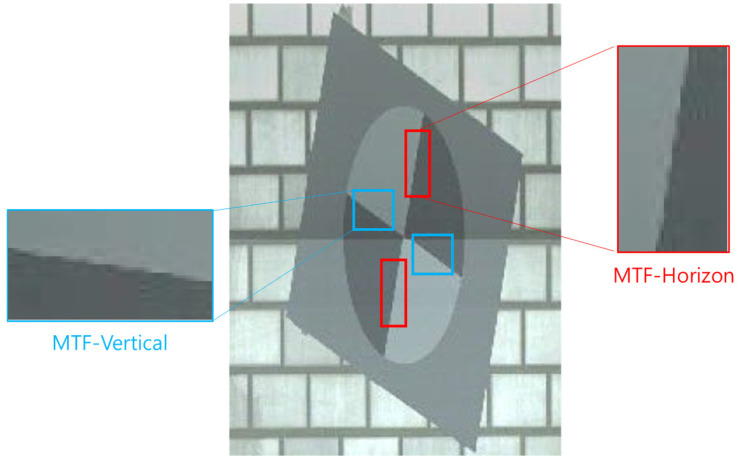
ROIs for horizontal (red) and vertical (blue) MTF analysis extracted from the SFRreg chart image.

**Figure 21 sensors-25-03804-f021:**
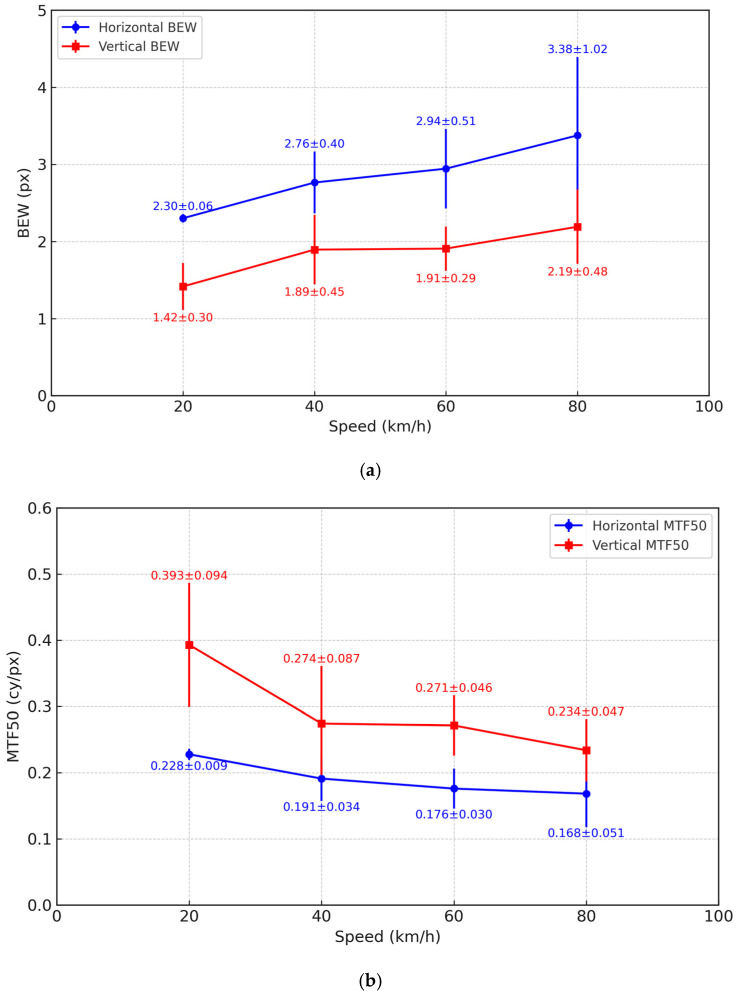
Motion blur characteristics of tunnel inspection images using an SFR chart. (**a**) Variation in horizontal and vertical BEW by speed. (**b**) Variation in horizontal and vertical MTF50 by speed.

**Table 1 sensors-25-03804-t001:** Test condition for capturing motion blur using a moving panel and area scan camera.

Panel Speed (km/h)	Shutter Speed (μs)	ISO	F-Number	FPS	Illuminance
0 10 30 50 70	500	640	2.8	100	15,000 lx 40,000 lx
	250	1250			
	100	1600			
	50	1600			

**Table 2 sensors-25-03804-t002:** Evaluation of motion-blurred images depending on the velocity of the moving panel and shutter speed at illuminance 15,000 lx.

Shutter Speed	IQA	0 km/h	10 km/h	30 km/h	50 km/h	70 km/h
500 μs	BEW_mean (pixels)	3.40	6.31	16.38	27.56	38.28
	BEW_std (pixels)	±0.38	±0.25	±0.09	±0.28	±0.31
	MTF50_mean (cy/px)	0.1566	0.0810	0.0375	0.0271	0.0227
	MTF50_std (cy/px)	±0.0155	±0.0208	±0.0324	±0.0352	±0.0364
250 μs	BEW_mean (pixels)	3.61	4.68	8.78	13.86	18.99
	BEW_std (pixels)	±0.41	±0.46	±0.12	±0.11	±0.29
	MTF50_mean (cy/px)	0.1485	0.1118	0.0606	0.0423	0.0338
	MTF50_std (cy/px)	±0.0133	±0.0154	±0.0262	±0.0311	±0.0334
100 μs	BEW_mean (pixels)	3.56	3.86	4.73	6.33	8.47
	BEW_std (pixels)	±0.33	±0.74	±0.33	±0.21	±0.21
	MTF50_mean (cy/px)	0.1505	0.1445	0.1069	0.0806	0.0626
	MTF50_std (cy/px)	±0.0114	±0.0253	±0.0153	±0.0209	±0.0257
50 μs	BEW_mean (pixels)	3.37	3.66	3.99	4.52	5.32
	BEW_std (pixels)	±0.26	±0.70	±0.5	±0.34	±0.43
	MTF50_mean (cy/px)	0.1581	0.1524	0.1337	0.1140	0.0955
	MTF50_std (cy/px)	±0.0128	±0.0253	±0.0162	±0.0143	±0.0182

**Table 3 sensors-25-03804-t003:** Evaluation of motion-blurred images depending on the velocity of the moving panel and shutter speed at illuminance 40,000 lx.

Shutter Speed	IQA	0 km/h	10 km/h	30 km/h	50 km/h	70 km/h
500 μs	BEW_mean (px)	3.23	6.07	15.18	25.19	35.37
	BEW_std (px)	±0.54	±0.34	±0.12	±0.51	±0.61
	MTF50_mean (cy/px)	0.1692	0.0853	0.0403	0.0295	0.0155
	MTF50_std (cy/px)	±0.0293	±0.0199	±0.0317	±0.0346	±0.0005
250 μs	BEW_mean (px)	3.52	4.39	8.29	12.83	17.44
	BEW_std (x)	±0.47	±0.35	±0.12	±0.18	±0.2
	MTF50_mean (cy/px)	0.1634	0.1172	0.0583	0.0374	0.0277
	MTF50_std (cy/px)	±0.0166	±0.0068	±0.0014	±0.0008	±0.0006
100 μs	BEW_mean (px)	3.49	3.61	4.76	6.47	8.37
	BEW_std (px)	±0.28	±0.31	±0.19	±0.15	±0.12
	MTF50_mean (cy/px)	0.1594	0.1456	0.1041	0.0743	0.0567
	MTF50_std (cy/px)	±0.0092	±0.0070	±0.0038	±0.0014	±0.0007
50 μs	BEW_mean (px)	3.42	3.71	3.95	4.43	5.30
	BEW_std (px)	±0.16	±0.35	±0.24	±0.15	±0.18
	MTF50_mean (cy/px)	0.1569	0.1432	0.1337	0.1135	0.0897
	MTF50_std (cy/px)	±0.0063	±0.0098	±0.0056	±0.0029	±0.0022

**Table 4 sensors-25-03804-t004:** Results of two-way ANOVA analyzing the effects of speed, shutter speed, and their interaction on image quality metrics (BEW and MTF50) under different illuminance levels (15,000 lx and 40,000 lx).

Illuminance	Metric	Source of Variation	DF	Sum of Squares	F-Value	*p*-Value
15,000 lx	BEW	Moving panel speed	4	17,036.88	29,998.22	<0.0001
		Shutter speed	3	18,673.68	43,840.35	<0.0001
		Interaction	12	14,139.28	8298.73	<0.0001
		Residual	580	82.35	-	<0.0001
	MTF50	Moving panel speed	4	0.82	365.26	<0.0001
		Shutter speed	3	0.39	233.39	<0.0001
		Interaction	12	0.11	16.48	<0.0001
		Residual	580	0.32	-	<0.0001
40,000 lx	BEW	Moving panel speed	4	14,555.51	36,821.60	<0.0001
		Shutter speed	3	15,158.78	51,130.25	<0.0001
		Interaction	12	11,647.52	9821.72	<0.0001
		Residual	580	57.32	-	<0.0001
	MTF50	Moving panel speed	4	1.04	1265.27	<0.0001
		Shutter speed	3	0.32	523.77	<0.0001
		Interaction	12	0.15	59.73	<0.0001
		Residual	580	0.12	-	<0.0001

Note: All *p*-values below 0.0001 are denoted as “<0.0001”. DF = degrees of freedom.

**Table 5 sensors-25-03804-t005:** Statistical significance (*p*-values) of BEW and MTF50 for illuminance change (15,000 lx vs. 40,000 lx) across different speed and shutter conditions.

Moving Panel Speed (km/h)	Shutter Speed (μs)	BEW *p*-Value	MTF50 *p*-Value
0	50	0.3372	0.644
	100	0.3564	0.0014 (*p* < 0.05)
	250	0.4386	0.0003 (*p* < 0.05)
	500	0.1645	0.0435 (*p* < 0.05)
10	50	0.7409	0.0715
	100	0.0944	0.8215
	250	0.0072 (*p* < 0.05)	0.0903
	500	0.003 (*p* < 0.05)	0.4239
30	50	0.7071	0.9949
	100	0.7594	0.3376
	250	0 (*p* < 0.05)	0.6427
	500	0 (*p* < 0.05)	0.733
50	50	0.1816	0.8499
	100	0.0029(*p* < 0.05)	0.1083
	250	0 (*p* < 0.05)	0.394
	500	0 (*p* < 0.05)	0.7915
70	50	0.7335	0.0965
	100	0.019 (*p* < 0.05)	0.214
	250	0 (*p* < 0.05)	0.3242
	500	0 (*p* < 0.05)	0.2914

**Table 6 sensors-25-03804-t006:** Descriptive statistics of BEW and MTF50 under two illuminance conditions (15,000 lx and 40,000 lx).

Metric	Illuminance	Q1 (25%)	Median (50%)	Q3 (75%)	Mean	Min	Max	Std. Dev.	Sample
BEW (pixels)	15,000 lx	3.94	5.13	10.17	9.48	2.65	39.14	9.13	600
	40,000 lx	3.79	5.1	9.59	8.95	2.67	36.64	8.32	600
MTF50 (cy/px)	15,000 lx	0.0534	0.1016	0.1411	0.096	0.013	0.1979	0.0524	600
	40,000 lx	0.0556	0.1005	0.141	0.0961	0.0145	0.2008	0.0521	600

Note: Q1 = 1st quartile, Q3 = 3rd quartile, Std. Dev. = standard deviation, N = number of samples. These values were calculated using raw measurement data from 600 images per illuminance level.

**Table 7 sensors-25-03804-t007:** Mean and standard deviation of BEW and MTF50 measured in both horizontal and vertical directions for different tunnel scanning speeds (20–80 km/h).

Direction	Speed (km/h)	Mean BEW (px)	Std. Dev.	Mean MTF50 (cy/px)	Std. Dev.
Horizontal	20	2.30	±0.06	0.228	±0.009
	40	2.76	±0.40	0.191	±0.034
	60	2.94	±0.51	0.176	±0.030
	80	3.38	±1.02	0.168	±0.051
Vertical	20	1.42	±0.30	0.393	±0.094
	40	1.89	±0.45	0.274	±0.087
	60	1.91	±0.29	0.271	±0.046
	80	2.19	±0.48	0.234	±0.047

## Data Availability

Dataset available on request from the authors.

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
