# Peer review of "Quality Assessment of High-Speed Motion Blur Images for Mobile Automated Tunnel Inspection"

_sensors, 2025, doi:10.3390/s25123804_

Round 1

Reviewer 1 Report

Comments and Suggestions for Authors

1. Limited Simulation of Motion Blur: The study focuses on horizontal translational blur but neglects vertical vibrations (common in real-world tunnel inspections due to uneven roads). Incorporating multi-directional or rotational motion blur would enhance practical relevance.

2. Insufficient Sample Size: Only 10 images per experimental condition (Figure 6) are analyzed, raising concerns about statistical significance. Expand the dataset (≥30 images per condition) and perform ANOVA to validate trends.

3. Unrealistic Illumination Levels: The selected illuminance (15,000–40,000 lx) far exceeds typical tunnel environments (500–2,000 lx). No justification is provided for these levels, and excessive reflections at 40,000 lx are noted but unquantified. Include low-illuminance tests (1,000–5,000 lx) and analyze ISO noise trade-offs.

4. Hardware Validation Gaps: Critical details of the high-speed translational panel (e.g., acceleration control, positional accuracy calibration) are missing. Provide laser-based stability verification to ensure motion consistency.

5.PSNR/SSIM vs. MTF/BEW Discrepancies: Lower PSNR/SSIM scores for high-shutter-speed images (despite reduced blur) are attributed to reduced contrast but lack validation. Correlate IQA metrics with crack detection accuracy using DL models (e.g., Faster R-CNN) to assess practical impact.

6.Statistical Reporting: Figures 10–11 lack error bars/confidence intervals, weakening reliability. Add p-values or statistical significance markers to support claims.

Author Response

Thank you for the opportunity to revise our manuscript. We appreciate the reviewers' insightful and constructive comments, which have been invaluable in enhancing the quality of our paper. We have carefully revised the manuscript in response to the reviewers' feedback and have also addressed several additional minor errors. The changes in our manuscript are marked in red.

Comments 1:  Limited Simulation of Motion Blur: The study focuses on horizontal translational blur but neglects vertical vibrations (common in real-world tunnel inspections due to uneven roads). Incorporating multi-directional or rotational motion blur would enhance practical relevance.

Response 1: Field validation was conducted using a mobile tunnel scanning system in operation. Motion blur (or sharpness) in the vertical direction was measured using SFR charts attached to tunnel walls.

Comments 2: Insufficient Sample Size: Only 10 images per experimental condition (Figure 6) are analyzed, raising concerns about statistical significance. Expand the dataset (≥30 images per condition) and perform ANOVA to validate trends.

Response 2: More than 30 image samples were used, and ANOVA was conducted.

Comments 3:Unrealistic Illumination Levels: The selected illuminance (15,000–40,000 lx) far exceeds typical tunnel environments (500–2,000 lx). No justification is provided for these levels, and excessive reflections at 40,000 lx are noted but unquantified. Include low-illuminance tests (1,000–5,000 lx) and analyze ISO noise trade-offs.

Response 3: It was not possible to capture images at a fast shutter speed of 50 μs under low-light conditions. A reference has been added to justify that illuminance above 20,000 lx is required.

Comments 4: Hardware Validation Gaps: Critical details of the high-speed translational panel (e.g., acceleration control, positional accuracy calibration) are missing. Provide laser-based stability verification to ensure motion consistency.

Response 4: A description of the control system that ensures consistent panel speed has been added.

Comments 5: PSNR/SSIM vs. MTF/BEW Discrepancies: Lower PSNR/SSIM scores for high-shutter-speed images (despite reduced blur) are attributed to reduced contrast but lack validation. Correlate IQA metrics with crack detection accuracy using DL models (e.g., Faster R-CNN) to assess practical impact.

Response 5: The PSNR/SSIM analysis section has been removed. In actual MTSS images acquired in the field, FR IQA is not feasible because reference (stationary) images are unavailable, making quality evaluation of in-motion images impossible.

The relationship between blurred images and DL-based crack detection models was mentioned in the Discussion as an area for future research. A separate paper on the impact of blur on DL model performance has already been written and submitted.

Comments 6: Statistical Reporting: Figures 10–11 lack error bars/confidence intervals, weakening reliability. Add p-values or statistical significance markers to support claims.

Response 6: This section has been removed.

Reviewer 2 Report

Comments and Suggestions for Authors

The main contribution of this study lies in the systematic investigation of motion blur assessment at high speeds under ISO-standard conditions, with a particular emphasis on tunnel automation detection. However, to improve the rigor and practical relevance of the research, it is important to clearly differentiate motion blur from other image degradation phenomena.

Detailed Comments:

  1. The manuscript should be revised to conform to the journal's formatting guidelines. It is recommended that the authors carefully review and adjust the layout before resubmission.

  2. The paper employs the ISO 12233 standard, which is designed for digital cameras and relies on test chart cards such as the ISO eSFR chart with uniformly focused textures. However, these charts cannot adequately simulate the complex optical interferences and non-uniform lighting reflections typical of tunnel surfaces.

  3. The blur in the study is simulated using translational panel movement. However, in real-world tunnel inspection scenarios, motion blur results from a combination of translational motion and vertical vibrations caused by vehicle dynamics. It is recommended to incorporate a high-speed vertical vibration module to replicate the suspension system's frequency response, along with an LED strobe control module to improve the fidelity of the simulation.

  4. The discussion section should be expanded to better articulate the limitations of the study.

  5. The conclusion focuses solely on quality assessment and does not provide quantitative relationships between image blur and the performance of downstream models (e.g., RetinaNet, DeepLab). For instance, the impact of motion blur on crack localization error (e.g., εâ‚“) when MTF50 falls below 0.05 is not addressed.

Comments on the Quality of English Language

The English could be improved to more clearly express the research.

Author Response

Thank you for the opportunity to revise our manuscript. We appreciate the reviewers' insightful and constructive comments, which have been invaluable in enhancing the quality of our paper. We have carefully revised the manuscript in response to the reviewers' feedback and have also addressed several additional minor errors. The changes in our manuscript are marked in red.

The main contribution of this study lies in the systematic investigation of motion blur assessment at high speeds under ISO-standard conditions, with a particular emphasis on tunnel automation detection. However, to improve the rigor and practical relevance of the research, it is important to clearly differentiate motion blur from other image degradation phenomena.

Comments 1: The manuscript should be revised to conform to the journal's formatting guidelines. It is recommended that the authors carefully review and adjust the layout before resubmission.

Response 1: The manuscript will be reformatted to comply with the journal’s formatting requirements.

Comments 2: The paper employs the ISO 12233 standard, which is designed for digital cameras and relies on test chart cards such as the ISO eSFR chart with uniformly focused textures. However, these charts cannot adequately simulate the complex optical interferences and non-uniform lighting reflections typical of tunnel surfaces.

Response 2: A field validation section has been added. The study conducted on-site validation using an actual mobile tunnel scanning system in operation. Motion blur (or sharpness) in the vertical direction was measured and analyzed using SFR charts attached to tunnel walls.

Comments 3: The blur in the study is simulated using translational panel movement. However, in real-world tunnel inspection scenarios, motion blur results from a combination of translational motion and vertical vibrations caused by vehicle dynamics. It is recommended to incorporate a high-speed vertical vibration module to replicate the suspension system's frequency response, along with an LED strobe control module to improve the fidelity of the simulation.

Response 3: The revised Discussion section discusses the inability to simulate vertical motion in the indoor environment. The field validation includes measurement and comparative analysis of vertical-direction MTF against the horizontal direction.

Comments 4: The discussion section should be expanded to better articulate the limitations of the study.

Response 4: The Discussion section has been newly written and expanded.

Comments 5: The conclusion focuses solely on quality assessment and does not provide quantitative relationships between image blur and the performance of downstream models (e.g., RetinaNet, DeepLab). For instance, the impact of motion blur on crack localization error (e.g., εâ‚“) when MTF50 falls below 0.05 is not addressed.

Response 5: The relationship between blurred images and crack detection deep learning models was mentioned in the Discussion as a topic for future research. A separate paper on the impact of blur on DL model performance has already been written and submitted.